# Nonlinear increase in seawater $^{87}$Sr/$^{86}$Sr in the Oligocene to early Miocene and implications for climate-sensitive weathering

Heather M. Stoll[1], Leopoldo D. Pena[2], Ivan Hernandez-Almeida[1], Jose Guitian[1*], Thomas Tanner [1], Heiko Pälike[3]

[1]Department of Earth Science, ETH Zurich, Zurich, 8092 Switzerland
[2]GRC Geociències Marines, Dept. Dinàmica de la Terra i de l'Oceà, Facultat de Ciències de la Terra, Universitat de Barcelona, Barcelona, 28080 Spain
[3]MARUM Centre for Marine Environmental Sciences, University of Bremen, Bremen, 28359 Germany

*Correspondence to*: Heather M. Stoll  (heather.stoll@erdw.ethz.ch)

*Present address: Centro de Investigación Mariña, Universidade de Vigo, GEOMA, Vigo, 36310, Spain

**Abstract.** The $^{87}$Sr/$^{86}$Sr of marine carbonates provides a key constraint on the balance of continental weathering and hydrothermal Sr fluxes to the ocean, and mid-Oligocene to mid-Miocene features the most rapid rates of increase in the $^{87}$Sr/$^{86}$Sr of the Cenozoic. Because previous records of the $^{87}$Sr/$^{86}$Sr increase with time were based on biostratigraphically defined age models in diverse locations, it was difficult to unambigiously distinguish m.y. scale variations in the rate of $^{87}$Sr/$^{86}$Sr change from variations in sedimentation rate.   In this study, we produce the first $^{87}$Sr/$^{86}$Sr results from an Oligocene to early Miocene site with a precise age model derived orbital tuning of high resolution benthic $\delta^{18}$O, at the Equatorial Pacific  Ocean Drilling Program (ODP) Site 1218.  Our new dataset resolves transient decreases in $^{87}$Sr/$^{86}$Sr, as well as periods of relative stasis.  These changes can be directly compared with the high resolution benthic $\delta^{18}$O in the same site. We find slowing of the rate of $^{87}$Sr/$^{86}$Sr increase coincides with the onset of Antarctic ice expansion at the beginning of the Mid-Oligocene Glacial Interval, and a rapid steeping in the $^{87}$Sr/$^{86}$Sr increase coincides with the benthic $\delta^{18}$O evidence for rapid ice retreat.  This pattern may reflect either northward shifts in the Intertropical Convergence Zone precipitation to areas of nonradiogenic bedrock, and/or lowered weathering fluxes from highly radiogenic glacial flours on Antarctic.  We additionally generate the first $^{87}$Sr/$^{86}$Sr data from ODP Site 1168 on the Tasman Rise and Integrated Ocean Drilling Program (IODP) Site 1406 of the Newfoundland Margin during the Oligocene to early Miocene to improve the precision of age correlation of these Northern Hemisphere and Southern Hemisphere mid-latitude sites, and to better estimate the duration of early Miocene hiatus and condensed sedimentation.

## 1 Introduction

The mid-Oligocene through the mid-Miocene features the fastest rate of change in seawater $^{87}$Sr/$^{86}$Sr of the Cenozoic, evidence of significant change in the balance of Sr sources to the ocean.  Although the precise

causes of the Cenozoic $^{87}Sr/^{86}Sr$ change remain under discussion, to first order the rise reflects an increase in the
supply of dissolved Sr sourced from weathering of older rocks of higher $^{87}Sr/^{86}Sr$, which are found on continents,
compared to the supply of dissolved Sr from rocks of lower $^{87}Sr/^{86}Sr$ characterizing submarine volcanic
weathering and subaerial weathering of young volcanic provinces (Palmer and Elderfield, 1985). This change in
balance of sources can be accomplished by one or more processes including decrease in rate of hydrothermal
weathering, increase in total continental weathering, or changes in the $^{87}Sr/^{86}Sr$ of continental weathering flux due
to either changes in the location of most intense weathering or changes in the composition and average age
(Peucker-Ehrenbrink and Fiske, 2019) of rocks exposed to weathering.
The Oligocene-early Miocene is a period of very rapid increase in $^{87}Sr/^{86}Sr$, with multiple possible drivers
including the unroofing of highly radiogenic source rocks in the Himalaya (Galy et al., 1996; Yang et al., 2022;
Myrow et al., 2015). Within the Oligocene-early Miocene period of rapid increase in $^{87}Sr/^{86}Sr$, some previous
studies have suggested the potential for 1-3 million year timescale variations in the rate of increase (Oslick et al.,
1994) and proposed that the liberation of Sr from silicate weathering may respond to changes in the production
and exposure of glacially floured rock on Antarctica (Miller et al., 1991; Oslick et al., 1994; Zachos et al., 1999).
However, the precision of estimates of the rate of change in $^{87}Sr/^{86}Sr$ are limited by the precision of the
independent age model in marine records. Where age model control points are of low resolution or low certainty,
changes in sedimentation rate may cause apparent variations in the rate of change in $^{87}Sr/^{86}Sr$, so that changes in
the rate of $^{87}Sr/^{86}Sr$ cannot be confidently inferred. To date, available $^{87}Sr/^{86}Sr$ data for the Oligocene and early
Miocene is derived from deep sea sediment cores featuring only biostratigraphically derived age models, whose
precision is limited by the biostratigraphic sampling resolution as well as the potential for diachroneity among
events. Precision on such age models can be limited by long distances between examined biostratigraphic points
in the core and the potential for diachroneity in the first occurrence or last occurrence of taxa in diverse locations,
and may feature uncertainties from 0.5 to 4 million years (Miller et al., 1988). Over the last decade,
astrochronology has emerged as a powerful independent chronometer, and the success of continuous coring and
splicing of deep ocean sediment cores has enabled the elaboration of precise independent age models based on
orbital tuning of high resolution benthic $\delta^{18}O$ records (Pälike et al., 2006; Liebrand et al., 2016; Westerhold et al.,
2020; De Vleeschouwer et al., 2017).
In this study, we seek to apply the independent orbitally tuned Oligocene chronology for two purposes.
First, we seek to evaluate the potential for dynamic changes in Sr sources by producing a $^{87}Sr/^{86}Sr$ record from a
site with an independent orbitally-resolved age model, Ocean Drilling Program (ODP) Site 1218 from the
Equatorial Pacific (Figure 1), for which original chronology (Pälike et al., 2006) was recently updated (Westerhold
et al., 2020). The very rapid rate of change in seawater $^{87}Sr/^{86}Sr$ also provides the opportunity for improved age
correlation among distal sites (Mcarthur et al., 2020). Therefore, our second objective is to improve the fidelity
of the age model for two further sites which currently lack an orbitally resolved age model, using existing reference
curves and the Site 1218 record as an additional reference. For this objective, we focus on North Atlantic
International Ocean Discovery Program Site 1406 (Newfoundland Margin) and Southern Ocean ODP Site 1168
(Tasman Rise), both emerging as important sites for Oligocene to early Miocene paleoceanographic studies (Scher
et al., 2015; Hoem et al., 2022; Hoem et al., 2021; Guitián and Stoll, 2021; Kim and Zhang, 2022; Egger et al.,
2018; Liu et al., 2018; Spray et al., 2019; Boyle et al., 2017). At site 1406, Sr isotope stratigraphy improves
constraints on the duration of an early Miocene hiatus (van Peer et al., 2017b; Norris et al., 2014). The Oligocene
to Early Miocene Southern Ocean paleogeography produced strong provincialism in many marine taxa from Site
1168, so synchroneity with global biostratigraphic datums is uncertain.  For paleoclimatic study, tuning to Site
1218 offers the advantage of providing a precise link with the complete benthic $\delta^{18}O$ record and therefore enabling
direct correlation to the highest resolution paleoclimatic record available for this time interval.
**Figure 1. Location of ODP 1218, IODP 1406, and ODP 1168 with paleogeography during the Oligocene-Miocene**
**Transition.  Reconstruction was made using the plate tectonic reconstruction service ODSN (www.odsn.de).**

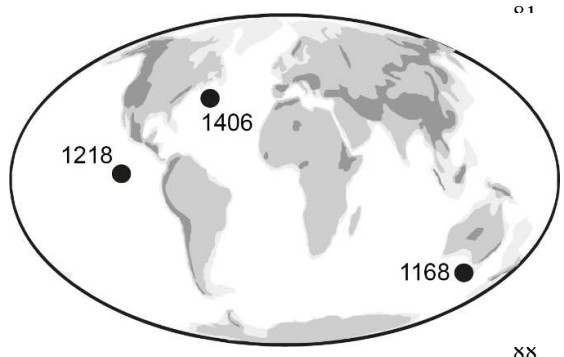

## 2 Sites and Methods

### 2.1 Sediments

Ocean Drilling Program (ODP)  Leg 199, Site

Site 1218, equatorial Pacific (8°53.378′N, 135°22.00′W,

4.8-km water depth) features a detailed astrochronologic

age model from benthic $\delta^{18}O$ originally spanning 22 to

25 Ma (Pälike et al., 2006). Subsequently,  continuous

tuning at precision of the 100 ky eccentricity cycle from
21.81 Ma through the lowermost Oligocene was generated on the CENOGRID timescale (Westerhold et al.,
2020). In Site 1218 we sought high resolution in the Middle Oligocene Glacial Interval (MOGI), previously
hypothesized to feature inflection points in the $^{87}Sr/^{86}Sr$ curve (Oslick et al., 1994). We targeted samples between
59.93 and 211.94 revised meter core depth (rmcd).  Due to the modest carbonate content of Site 1218, not all
targeted sample intervals contained sufficient foraminifera for analysis.  We have picked >2 mg of mixed species
of planktonic or mixed species of benthic foraminifera, depending on the abundance in each sample. From some
samples, populations of both benthic and planktic forams could be procured and we report the averaged $^{87}Sr/^{86}Sr$
ratio for the two populations.

Integrated Ocean Drilling Program (IODP) Expedition 342 recovered Paleogene to Neogene sedimentary
sequences in contourite drift deposits off the coast of Newfoundland in the Northwestern Atlantic.  Here we focus
on Site 1406 (40°21.0′N, 51°39.0′W; 3814 mbsl) with samples dominantly from Hole A, but including a few
samples from Holes B and C. The composite depth scale for the site (CCSF-A) is based on physical properties
and trace element ratios from XRF Scanner (van Peer et al., 2017b) and is under revision as further benthic
foraminifera and fine fraction stable isotope data are produced.  Consequently, where samples from all sites are
plotted, we illustrate on the composite CCSF-A scale but where exclusively data from Hole A are presented, we
illustrate depth on CSF-A scale as this latter scale will not be revised; both depth scales are provided in data tables.
Based on available biostratrigraphy and previous age models (Norris et al., 2014)  (van Peer et al., 2017b), we
sought samples spanning age range 17 to 30 Ma, represented by depths from 23.9 to 200 m on the CCSF-A depth
scale.  In the Southern Hemisphere, ODP Site 1168 was drilled offshore of the Australian plate at the western
margin of Tasmania, at 43° 36.57'S and 139 144° 24.76'E, and 2463m water depth.  This sequence is within a
graben-developed basin with sediment accumulation since the latest Eocene (Exon et al., 2004). Based on
available biostratigraphy (Stickley et al., 2004), we selected samples from Hole A, spanning the 16 to 27 Ma
interval, representing sediments from 278 to 562 m depth on the mbsf depth scale used for ODP sites of this
generation. Mixed planktonic foraminifera were picked for both sites 1406 and 1168.

## 2.2 Analytical

Strontium isotope ratios ($^{87}Sr/^{86}Sr$) were measured on ~2 mg of cleaned foraminifera carbonates. Foraminifera samples were crushed open under binocular inspection and the fragments were rinsed several times in MilliQ water, methanol and ultrasonicated to remove detrital contaminants (Pena et al., 2005). Each sample was treated individually to ensure that sufficient rinsing steps were applied. Cleaned fragments were dissolved in dilute double distilled nitric acid, and the resulting solution centrifuged at medium speed for 20 minutes to remove any potential detrital material left in the samples. The supernatant was transferred to clean Savillex-PFA beakers and Sr was chemically separated from sample matrix and interferring Rb using Triskem Sr-Spec resin through column chromatography procedures at the LIRA ultra-clean laboratory (Universitat de Barcelona).

Following sample purification, Sr isotope ratios were determined by multicollector inductively coupled mass spectrometry on a Nu Instruments (Wrexham, UK) Plasma 3 MC-ICPMS at the University of Barcelona (CciT-UB). For the determination of the $^{87}Sr/^{86}Sr$ isotope ratios, the contribution of $^{87}Rb$ to the $^{87}Sr$ signal was corrected from the measurement of the $^{85}Rb$ signal, assuming a $^{87}Sr/^{85}Sr$ ratio of 0.38562. The $^{86}Kr$ interference on $^{86}Sr$, caused by impurities in the argon gas, was also corrected by measuring the $^{83}Kr$ signal, and assuming a $^{83}Kr/^{86}Kr$ value of 0.66453. $^{87}Sr/^{86}Sr$ ratios were normalized for instrumental mass bias to $^{86}Sr/^{88}Sr = 0.1194$. Instrumental drift was corrected by sample-standard bracketing (SSB) using NBS987 = 0.710249 as the primary standard with matching standard and sample Sr concentrations. External analytical reproducibility during the session was $\pm 0.000018$ ($2\sigma$, n=19). Procedural blanks are routinely measured at every analytical session. Typical procedural Sr blanks (including sample cleaning, purification and analysis) are $369 \pm 264$ pg, n=12, $1\sigma$. Blanks are systematically corrected for every measurement and the effect of the correction is in the sixth decimal place of the $^{87}Sr/^{86}Sr$ ratios, well below the external reproducibility of the analytical method (5th decimal place). Values are normalized to SRM 987 using $^{86}Sr/^{86}Sr$ of 0.1194 and $^{87}Sr/^{86}Sr$ of 0.710249. This is identical to the normalization of (Mcarthur et al., 2020) using $^{87}Sr/^{86}Sr$ 0.709174 for modern marine-Sr (EN-1 and similar), equivalent to 0.710248 for SRM(NIST) 987.

## 3 Results

Our new data from Site 1218 on the Cenogrid age model (Figure 2, Table 1), reveal a similar long term amplitude and rate of rise in $^{87}Sr/^{86}Sr$ as previously reported data on biostratigraphically constrained age models (Mcarthur et al., 2020). However, the new data reveal a 1 m.y. duration period of negligible $^{87}Sr/^{86}Sr$ rise (27-28 Ma) and local reversals in the overall trend of increasing $^{87}Sr/^{86}Sr$ during the Middle Oligocene Glacial Interval (MOGI) and at the Oligo-Miocene transition. The new data also reveal several intervals of especially abrupt increase in $^{87}Sr/^{86}Sr$ within and at the end of the MOGI.

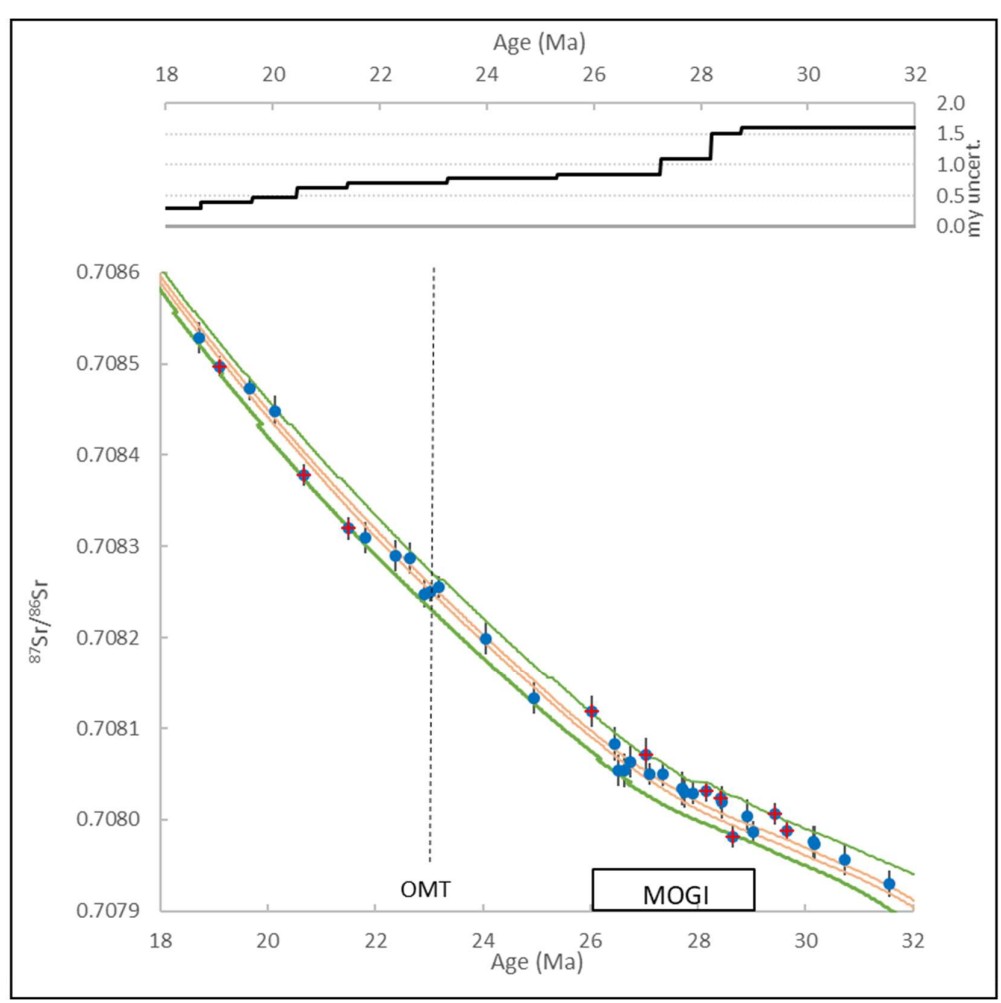

147

**Figure 2:** **$^{87}$Sr/$^{86}$Sr results from Site 1218 (blue circles with 2σ analytical uncertainty shown). The orange lines show the upper and lower bounds of the LOESS fit of biostratigraphically defined $^{87}$Sr/$^{86}$Sr (Mcarthur et al., 2020). Samples falling outside the biostratigraphically defined long term curve are highlighted in red. The green lines illustrate expanded age bounds for LOESS fit of biostratigraphically constrained age models (Mcarthur et al., 2020). Upper panel illustrates the width of the age uncertainty of the expanded bounds. The Middle Oligocene Glacial Interval (MOGI) from 29 to 26 Ma is labelled, as is the Oligocene-Miocene Transition (OMT). We highlight this duration of MOGI on the basis of the 1218 benthic δ$^{18}$O record as indicated in Figure 4; it is slightly longer than the 28 to 26.3 Ma MOGI defined by (Liebrand et al., 2016; Liebrand et al., 2017)**


In Site U1406, a prominent reversal in the $^{87}$Sr/$^{86}$Sr rise is observed between 48.7 and 45.4 m (Figure 3a,
depths described on the CCSF-A scale). A significant jump in $^{87}$Sr/$^{86}$Sr suggests an appreciable hiatus between
33.3 m and 34.7 m.  The abrupt change in $^{87}$Sr/$^{86}$Sr between 176.2 and 170.1 may also indicate a hiatus or
significantly condensed interval.  In Site 1168 (Figure 3b), a prominent reversal in the $^{87}$Sr/$^{86}$Sr rise occurs between
538.2 and 527.8 m CSF-A.

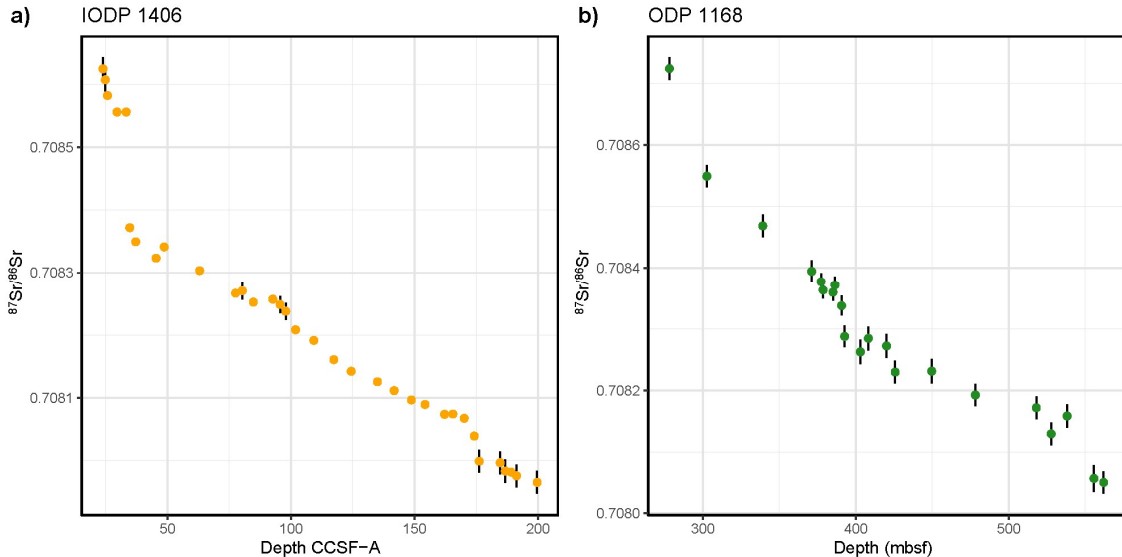


**Figure 3:** $^{87}$Sr/$^{86}$Sr results from a) U1406 and b) 1168. Vertical error bars indicate 2σ analytical uncertainty where it
exceeds the size of the plotted symbol.

## 4 Discussion

### 4.1 Variation in the rate of change of $^{87}$Sr/$^{86}$Sr in Site 1218

The steep long term Oligocene to early Miocene increase in $^{87}$Sr/$^{86}$Sr is long recognized and variably
attributed to exhumation of readily weathered radiogenic bedrock during the Himalayan orogeny (Krishnaswami
et al., 1992; Raymo et al., 1988), or to accelerated weathering of radiogenic bedrock in Antarctica with the onset
of its glaciation (Miller et al., 1991). The cause of this long term increase is beyond the scope of this study and
our focus is on the variability in the rate of increase within the Oligocene to early Miocene.

Significantly, the precise independent chronology of Site 1218 confirms that the long term Oligocene
and early Miocene increase in $^{87}$Sr/$^{86}$Sr is punctuated by significant structure, including reversals, periods of
negligible $^{87}$Sr/$^{86}$Sr increase, as well as more abrupt increases in $^{87}$Sr/$^{86}$Sr . Reversals beyond analytical
uncertainty are also seen in published high resolution Mid-Oligocene $^{87}$Sr/$^{86}$Sr records from both ODP Site 522
(Reilly et al., 2002) and ODP Site 689B (Mead and Hodell, 1995), suggesting that they are a robust feature of the
Oligocene ocean Sr cycle. The uncertainty in absolute chronology complicates the inference of m.y. scale periods
of stasis or abrupt $^{87}$Sr/$^{86}$Sr increase in previously published records with biostratigraphic age models. To further
evaluate how the rate of change of $^{87}$Sr/$^{86}$Sr deviates from a monotonic increase from 32 to 18 Ma, we generate
a smoothed fit to the data based on local linear regression model (Figure 4). In the model, local regressions were
based on 3 to 6 consecutive samples and age range of at least 0.25 Ma, with the exception of a single shorter span
of only 0.18 Ma at 26 Ma. To estimate the rate of change, we illustrate the derivative of this smoothed fit, as well
as the slope and its uncertainty for each linear segment (Figure 4b). This analysis illustrates periods of both more
rapid increase as well as slowed $^{87}$Sr/$^{86}$Sr increase or a decrease.

In Site 1218, appreciably lower rates of $^{87}$Sr/$^{86}$Sr increase (or even $^{87}$Sr/$^{86}$Sr decrease) occur centered at
29 Ma and 26.8 Ma during the MOGI, and at 23 Ma during the OMT (Figure 4). Each of these periods is followed

by a large acceleration of $^{87}Sr/^{86}Sr$ increase. Our new data provide the most precise comparison between $^{87}Sr/^{86}Sr$
and the benthic $\delta^{18}O$ record of deep sea temperature and ice volume because the records derive from the same
deep sea sediment archive (without correlation uncertainty) and the benthic $\delta^{18}O$ record is very high resolution,
without aliasing which can occur in records sampled at resolution comparable or greater than periods of orbital
variation. The earliest slowing in the rate of $^{87}Sr/^{86}Sr$ increase and even decrease in $^{87}Sr/^{86}Sr$ which we resolve
(centered at 29 Ma) coincides with the onset of heavier average benthic $\delta^{18}O$ demarcating the Mid-Oligocene
Glacial interval (MOGI), and the recovery of rapid $^{87}Sr/^{86}Sr$ increase coincides with a shift towards more negative
benthic $\delta^{18}O$ (ice volume decrease and/ or deep sea warming). The return to more intense glaciation from 28 to
26.8 Ma yields a decrease in $^{87}Sr/^{86}Sr$. The subsequent acceleration of $^{87}Sr/^{86}Sr$ increase at 26.5 Ma coincides
with the onset of the negative shift in benthic $\delta^{18}O$ marking the end of the MOGI with ice volume decrease and/
or deep sea warming. The reduction of $^{87}Sr/^{86}Sr$ increase (or $^{87}Sr/^{86}Sr$ decrease) at the OMT coincides with an
intense glacial phase, and subsequently a consequent acceleration of $^{87}Sr/^{86}Sr$ increase at the end of the glacial
phase. This event may coincide with the post-OMT acceleration previously defined  as 22.4 Ma on the Cande and
Kent (Cande and Kent, 1992) timescale (Oslick et al., 1994). We are unable to evaluate if there are similar <0.5
Ma variations in the rate of $^{87}Sr/^{86}Sr$ change between 26 and 23 Ma as our sample resolution is not high enough
in this interval. The main changes in slope are significant at the 68% CI (1s) level, but an increase in the sample
resolution and number of data points would be needed to confidently distinguish many of these differences at the
95% CI (2s) level.

207          Variations in the isotopic composition of Sr inputs on timescales of $10^5$ yr  are not expected to reflect

changes in ocean crustal production rate and hydrothermal flux, nor significant changes in the composition of
bedrock exposed on continents. Therefore, such changes are suggestive of change in either the intensity of
continental weathering relative to hydrothermal sources or changes in the locus of most intense continental
weathering among continental sources of contrasting $^{87}Sr/^{86}Sr$. For example, a short term relative increase in
weathering intensity in areas underlain by younger average bedrock compared to older average bedrock would
lead to a decreased $^{87}Sr/^{86}Sr$ of the riverine Sr flux and the marine reservoir. Alternatively, a short term decrease
in the intensity of weathering and thereby in the continental Sr flux (higher $^{87}Sr/^{86}Sr$  than the hydrothermal flux)
could also lead to a decreased marine $^{87}Sr/^{86}Sr$. In either case, the long residence time of Sr in the ocean would
result in lags between onset of elevated fluxes and peak response in ocean chemistry and would cause significant
attenuation of the time-varying input signal. An example of the phasing and amplitude variation in the $^{87}Sr/^{86}Sr$
of the Sr influx which could yield the observed trends in marine $^{87}Sr/^{86}Sr$ is illustrated in Supplementary Figure
1. for a sample residence time of 2.5 million years as suggested by (Hodell et al., 1990) Hodell et al., 1990. A
shorter residence time has been proposed for the Oligocene by (Paytan et al., 2021); for a shorter residence time,
a less extreme forcing would be required to simulate our observations. We caution that because the Sr isotopic
system of the Oligocene to Early Miocene is underconstrained, the observations of oceanic $^{87}Sr/^{86}Sr$ do not provide
a unique solution for the variation in fluxes and/or their isotopic composition.

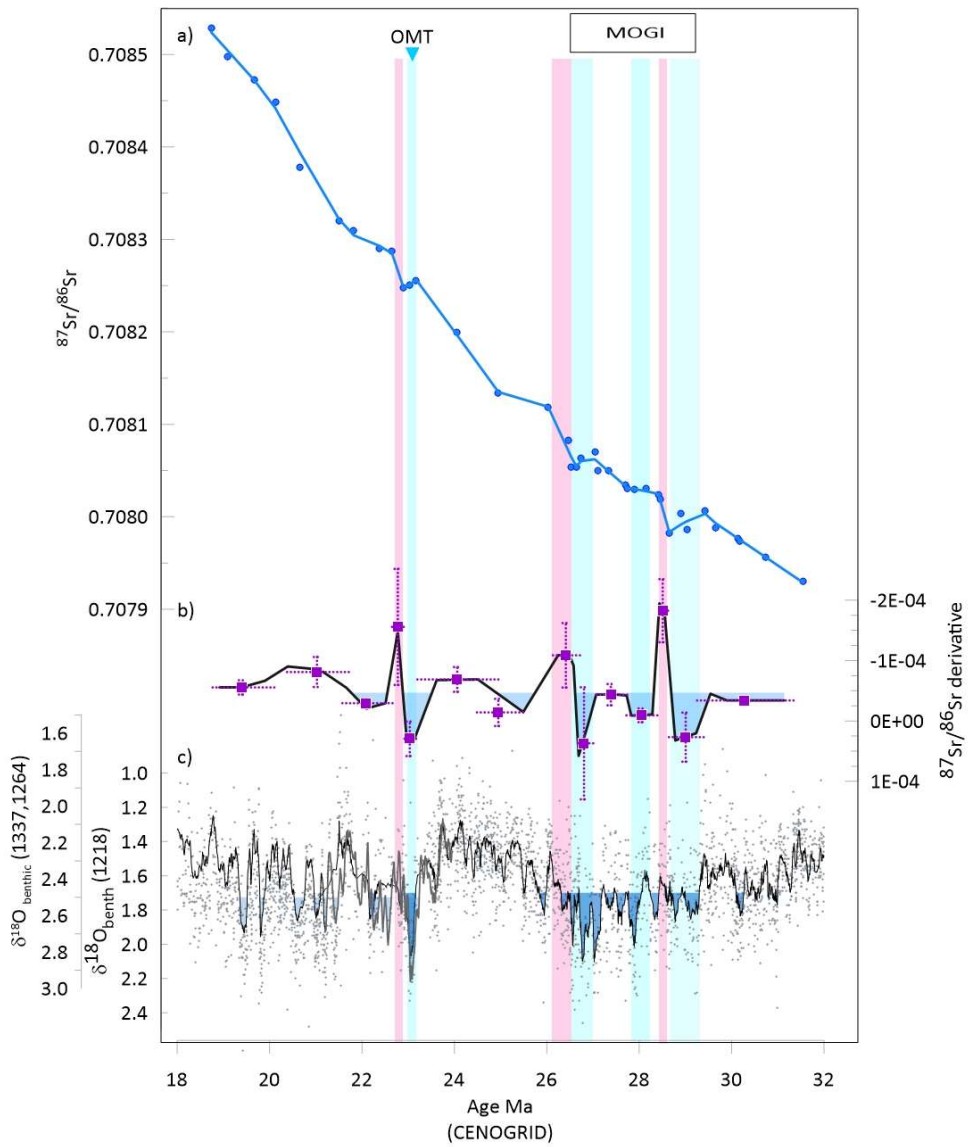


**Figure 4. a) Measured Site 1218 $^{87}Sr/^{86}Sr$ (symbols) and the smoothed fit from local linear regression (blue line). b) the derivative of the smoothed fit (black line) and the slope of each linear segment (purple square), together with the uncertainty on the slope (vertical error bar, 68% confidence interval) and the age range of the local linear fit (horizontal bar). Shading indicates sectors in which $^{87}Sr/^{86}Sr$ rises more slowly than the average rate over 32 to 18 Ma. c) Benthic $\delta^{18}O$ measurements (gray points) and lines showing 20 point running mean. From Site 1218 (black line from 21.46 to 32 Ma) (Pälike et al., 2006) and from the Cenozoic reference splice derived from U1337 (Holbourn et al., 2015) and ODP Site 1264 (Westerhold et al., 2020) both scaled as in (Westerhold et al., 2020), as gray line from 21.2 to 24 Ma when overlapping with 1219 record, and black line from 18 to 21.2 Ma. All data are plotted on the orbitally tuned CENOGRID timescale (Westerhold et al., 2020). Blue shading highlights intervals with benthic $\delta^{18}O$ more positive than 1.7 ‰ in Site 1218 and 2.5 ‰ in Cenozoic reference splice 1264 and U1337. The Middle Oligocene Glacial Interval (MOGI) from 29 to 26 Ma is labelled, as is the Oligocene-Miocene Transition (OMT). Vertical blue and pink lines highlight intervals of slower and more rapid rate of change in $^{87}Sr/^{86}Sr$, respectively.**

The coincidence of periods of slowed $^{87}Sr/^{86}Sr$ and the onset of glacial advance on Antarctica evidenced in benthic $\delta^{18}O$ suggests a climate control on the variations in the continental Sr flux on $10^5$ yr timescales from one or both of these mechanisms. Changes in the location of intense rainfall, such as shift in the polar front or Intertropical Convergence Zone (ITCZ), could alter the locus of most intense weathering. Potentially, episodes of Antarctic glacial expansion could cause an equatorward movement of the SH westerlies and associated rainfall band, or could cause a mean ITCZ shift toward the northern Hemisphere. However, climatically-driven changes

in the position of main heavy rainfall belts such as ITCZ is usually limited to < 10 degrees latitude and may be longitudinally variable (Atwood et al., 2020). A movement of precipitation belts would have a significant consequence on global riverine $^{87}Sr/^{86}Sr$ only in cases of fortuitous distribution of bedrock of widely different ages across the length scale of ITCZ movement. If a northward shift of the mean ITCZ significantly increased the Sr flux from a region of nonradiogenic Sr, the marine $^{87}Sr/^{86}Sr$ could experience a transient decrease. One potential such configuration could be the exposure of highly weatherable nonradiogenic rocks of the Deccan volcanic series of India and the Ethiopian Traps, located just north of the equator in the late Oligocene (Kent and Muttoni, 2013).

It has also been proposed that glaciation can affect the weatherability of bedrock. Generally, highest riverine dissolved Sr fluxes are produced from reactive young volcanic rock, as well as soluble carbonates, but the mechanical flouring of less reactive rock types by glacial erosion can significantly increase their weatherability and Sr contribution to the ocean. It has been suggested that weathering intensity of the Antarctic craton may have evolved over the late Eocene through Oligocene, as glacial flouring of Antarctic bedrock increased the weatherability of this continental Sr source (Miller et al., 1991; Oslick et al., 1994), contributing to the rise in ocean $^{87}Sr/^{86}Sr$. Intermittent glaciation, characterized by significant changes in the spatial extent of ice coverage, may alternately generate highly weatherable fine grained silicates in a subglacial weathering-limited environment (low continental Sr fluxes) and then expose them to subaerial conditions of enhanced chemical weathering (high continental Sr flux). On previous biostratigraphic age models, apparent accelerations in the rate of $^{87}Sr/^{86}Sr$ increase at 32, 28, 22.4, 19.5, and 16.5 Ma (on the Cande and Kent timescale) occur 1 m.y after deglaciation midpoint inferred from benthic $\delta^{18}O$ maxima in ODP 747 (Oslick et al., 1994). This was interpreted to result from the deglacial exposure which may have contributed to a transient increase in flux of radiogenic Sr to the ocean. With higher resolution benthic $\delta^{18}O$ from 1218, we resolve more rapid responses of the $^{87}Sr/^{86}Sr$ ratio to several deglaciation phases.

East Antarctica is inferred to be underlain dominantly by Proterozoic and Archean bedrock (Kirkham et al., 1995). Exposed bedrock in East Antarctica is dominated by Archean and Proterozoic metamorphic rocks, with Paleozoic igneous and sedimentary rocks additionally exposed in the Transantarctic mountains (Licht and Hemming, 2017). Although the Sr isotopic composition of bedrock in Antarctica can be measured directly only in current exposures in the Transantarctic mountains and coastal areas, crustal rocks at the perimeters of major ice sheets may represent a major source of the sediment arriving at the margin and therefore weatherable during retreat (Farmer et al., 2003). Because erosion rates are highest at the perimeter of Antarctic ice sheet (Jamieson et al., 2010), the mapped bedrock in coastal areas may provide a reasonable representation of the source of sediment arriving to the glacial margin and weatherable during retreat. Additionally, the fine grained component of LGM tills exposed in the Ross Sea embayment provide constraints on modern underlying composition of present erosion (Farmer et al., 2006). Present till composition includes very radiogenic compositions up to 0.740 attributed to erosion of the Neoproterozoic Beardsmore Formation, and compositions in the range of 0.720 to 0.735 typical of 500 Ma Granite Harbor Intursive rocks exposed in southern part of Transantarctic Mountains and the Wilson terrane Proterozoic gneisses (Farmer et al., 2006). However, a caveat is that the currently exposed bedrock may be older than that exposed in the Oligocene due to denudation, and younger, less radiogenic bedrocks may have contributed more to glacial flouring in the Oligocene, making global fluxes less sensitive to the Antarctic weathering regime.

**4.2 Sr isotope constraints on age models of Site U1406 and Site 1168**


Previous approaches for Sr isotope stratigraphy for the Cenozoic have inferred a continuous rise in
$^{87}Sr/^{86}Sr$. The data from Site 1218 suggest several intervals with a negligible rate of rise and/or reversals. In the
interval from 28 to 30 Ma, 5 of our 9 samples feature $^{87}Sr/^{86}Sr$ ratios whose analytical 95% CI fall outside of the
bounds of the reference $^{87}Sr/^{86}Sr$ curve of that age generated from biostratigraphically constrained age models
(Mcarthur et al., 2020). For these intervals, particularly during the MOGI, age assignments from Sr isotope
stratigraphy have a higher uncertainty than previously inferred. In the interval from 28 to 30 Ma, the deviation
between the CENOGRID age and the reference curve ranges from 1.1 Ma older than the reference curve to 0.7
Ma younger, a significantly wider uncertainty than the +/- 120 to 180 ky uncertainty predicted for the reference
curve. In the early Miocene, between 21.7 and 19.4 Ma, a number of our Site 1218 CENOGRID ages also deviate
from the ages from the reference curve, by 0.18 to 0.42 Ma younger, a greater uncertainty than the +/- 70 to 50 ky
reported for the reference curve. Consequently, in deriving ages for Site U1406 and Site 1168 on the CENOGRID
scale from $^{87}Sr/^{86}Sr$, we expand the bounds of the age uncertainties from (Mcarthur et al., 2020) to encompass
the Site 1218 data (Figure 2, green bounds). The width of the resulting age uncertainty therefore ranges from 300
ky in the early Miocene to 1.6 My in the early-mid Oligocene.
In addition to these greater uncertainties, stratigraphic constraints prohibit reversals in ages where there
is no independent evidence for reworking or sediment disturbance. Our Site 1218 data indicate that reversals in
$^{87}Sr/^{86}Sr$ are certain within the time interval of 29-26 Ma, and likely at the OMT. Thus, in estimating ages for
1168 and U1406 based on $^{87}Sr/^{86}Sr$, we assign an initial age based on (McArthur et al 2020) but adjust the age to
preserve stratigraphic relationships (eg no age reversals in our age assignments). The detailed age models are
shown in Tables 2 and Table 3.

**4.2.1. Site U1406**


A condensed interval and hiatus have been recognized in the Oligocene to early Miocene sediments of
U1406 on the basis of bio- and magnetostratigraphy (Figure 5 ) (Norris et al., 2014; van Peer et al., 2017a).
Because of the slow rate of the Site U1406 $^{87}Sr/^{86}Sr$ change and reversals during the MOGI, the U1406 $^{87}Sr/^{86}Sr$
cannot precisely pinpoint the duration of the hiatus or condensed interval between 153 and 149 m (CSF-A) (Figure
5). The condensed interval could contain 2 m.y. (28.45 to 26.58 Ma) or < 1 m.y. (27.3 to 26.58). On the other
hand, the early Miocene condensed interval between 27 and 25 m (CSF-A) is constrained to represent 2 m.y.
Sustained high sedimentation rates are confirmed between 21 and 26 Ma. Between 26.4 and 21 Ma, the $^{87}Sr/^{86}Sr$
age model is in close agreement with that devised from magnetostratigraphy (Van Peer et al., 2017a). The Site
1406 $^{87}Sr/^{86}Sr$ data indicate that the uppermost 25 m of sediment in U1406, difficult to date due to sparse
biostratigraphic markers, was likely deposited between 18.5 and 17.6 Ma.

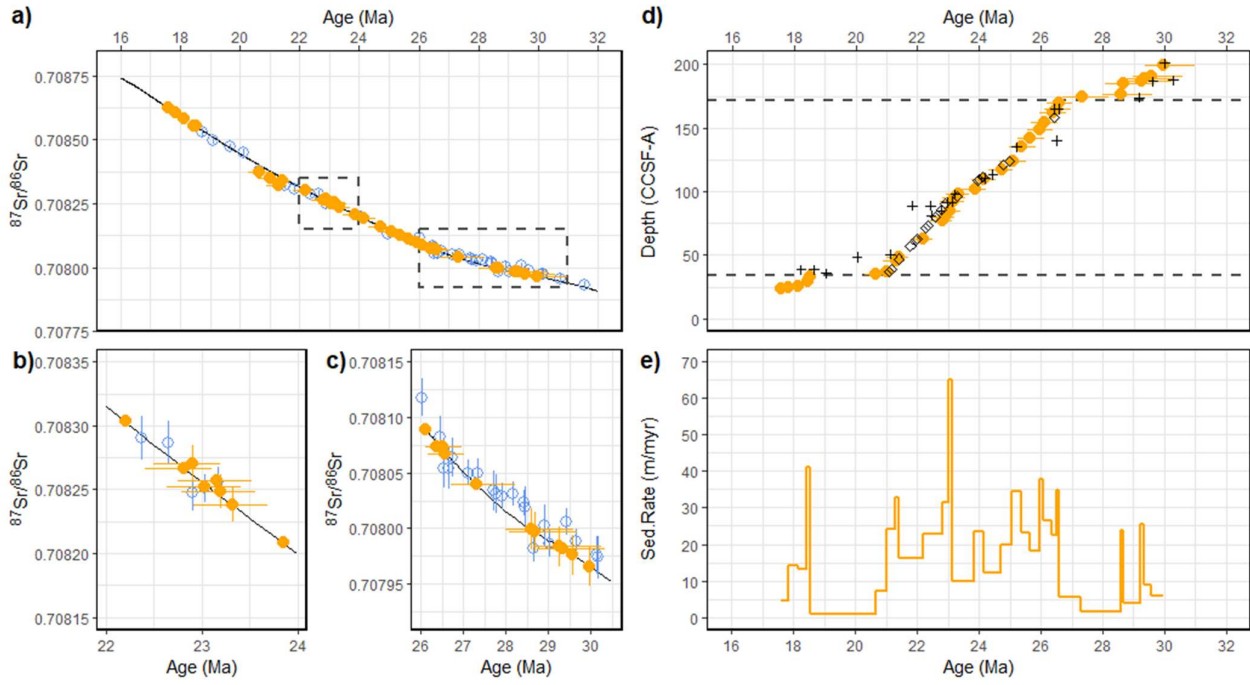


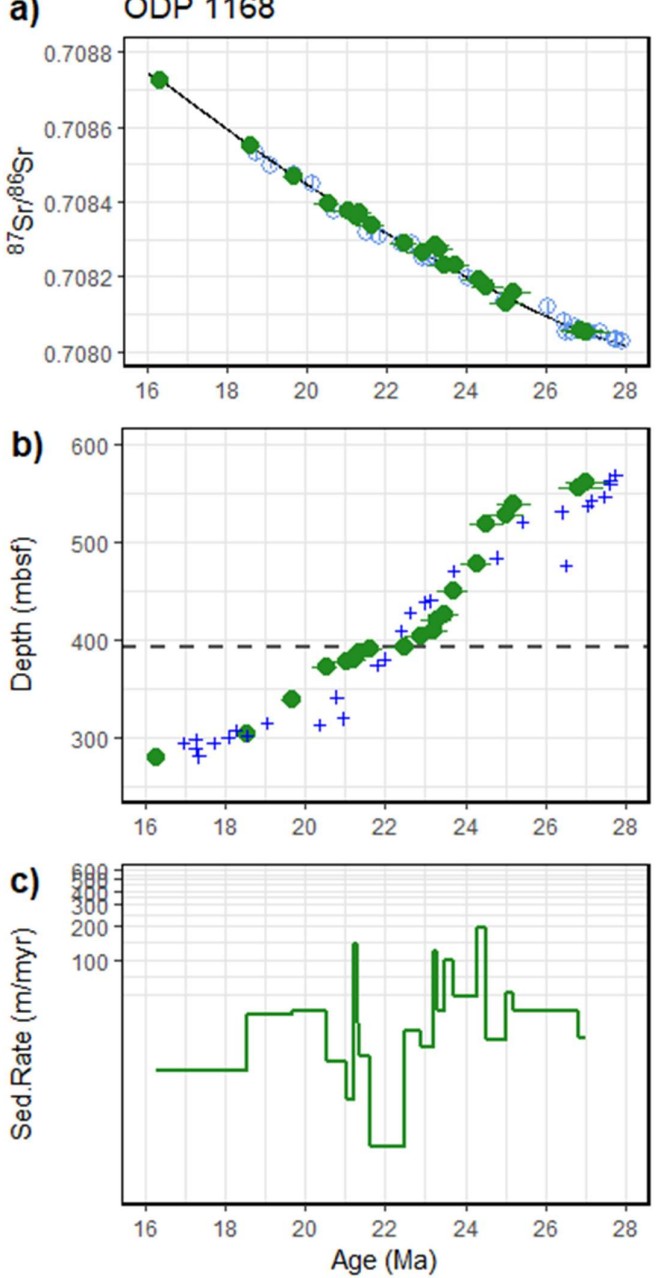

**Figure 5. a) Site U1406 <sup>87</sup>Sr/<sup>86</sup>Sr data (orange) vs age (CENOGRID scale) assigned here; also shown are the Site 1218 new data (blue) and the GTS LOESS curve (Mcarthur et al., 2020). b. and c. show insets. For Site U1406, horizontal lines indicate the uncertainty in the age assignments. d) Age depth plot for U1406A on the CSF-A scale. Orange circles denote the ages from <sup>87</sup>Sr/<sup>86</sup>Sr compared to previous biostratigraphy tiepoints (crosses; (Norris et al., 2014)) and magnetostratigraphy (open diamonds) (van Peer et al., 2017a). Horizontal dashed lines delimit the strongly condensed intervals; e) the inferred sedimentation rate.**

### 4.2.2. Site 1168

This first $^{87}$Sr/$^{86}$Sr stratigraphy for Site 1168 implies significant differences in inferred ages compared to existing biostratigraphy, including significantly higher sedimentation rates in the early Miocene (20.5 to 18.5 Ma), but comparably slower sedimentation rates between 21.6 and 22.5 Ma just after the OMT (Figure 6). This is slightly earlier than the early Miocene hiatus found in many deep sea sedimentary records between 19 to 20 Ma; however, in other deep sea records the precise timing of early Miocene depositional gaps is not yet resolved and could coincide with the condensed interval in 1168 (Sibert and Rubin, 2021). Age assignments remain less precise in the middle Oligocene (25 to 27 Ma) due to the low rate of change and reversals in $^{87}$Sr/$^{86}$Sr during this time interval, as well as the


current low $^{87}$Sr/$^{86}$Sr sample coverage.

**Figure 6: a) Site 1168 <sup>87</sup>Sr/<sup>86</sup>Sr data (green) vs age (CENOGRID scale) assigned here; also shown are the Site 1218 new**
**data (blue open symbols) and the GTS LOESS curve (Mcarthur et al., 2020) b). Age depth plot for Site 1168. Green**
**diamonds denote the ages from <sup>87</sup>Sr/<sup>86</sup>Sr compared to previous biostratigraphy derived tie points (crosses). c)**
**Sedimentation rate implied by the <sup>87</sup>Sr/<sup>86</sup>Sr age model. The horizontal dashed line in b) highlights the period of**
**significantly slowed sedimentation of the Early Miocene.**

## 5 Conclusions

The $^{87}$Sr/$^{86}$Sr record from the astrochronologically dated Site 1218 provides the opportunity to assess ~1 m.y. variations in the Sr flux to the ocean during a period of dynamic Antarctic cryosphere evolution. Our dataset resolves relationships between the locus and/or intensity of continental weathering and phases of Antarctic glaciation. Overall, the data suggest that the major changes in mid-Oligocene high latitude climate –particularly the onset and end of the MOGI – do exhibit a close coupling between seawater $^{87}$Sr/$^{86}$Sr and benthic $\delta^{18}$O. During periods of expanded ice coverage on Antarctica such as the MOGI, then, our data are consistent with either northward shifts in the ITCZ precipitation to areas of nonradiogenic bedrock, and/or lowered weathering fluxes from highly radiogenic glacial flours on Antarctic. Future, higher resolution sampling is required to further evaluate the significance of such changes. Additionally, the new $^{87}$Sr/$^{86}$Sr record from sites 1168 and U1406 improve the precision of age correlation of these Northern Hemisphere and Southern Hemisphere mid-latitude sites with each other and with high resolution benthic $\delta^{18}$O records aligned to the CENOGRID chronology.

## Competing interests

The contact author has declared that none of the authors has any competing interests.

## Acknowledgments

This study was supported by the Swiss National Science Foundation (Award 200021_182070 to Heather M. Stoll). We thank Romain Alosius for assistance picking foraminifera, and Laura Arnold for initial evaluation of foraminifera preservation. We gratefully acknowledge the Ocean Drilling Program and International Ocean Discovery Program for providing the samples used in this study.

## Author contributions

The study was conceived by HMS. Samples were selected by HMS with advice from HP. Foraminifera were prepared by JG, IHA, and TT. Sr isotope analyses were completed by LDP. Interpretation was completed by HMS and figures were prepared by HMS and JG. The manuscript was written by HMS with input from all authors.


**Table 1. ODP identifiers, CENOGRID age, and [87]Sr/[86]Sr data for Site 1218.**

| lab ID | Exp | Site | Hole | Core | Core Type | Section | Section Half | Top Interval (cm) | Bottom Interval (cm) | depth (rmcd, m) | age CENOGRID (Ma) | [87]Sr/[86]Sr | Internal SE (2σ) |
|---|---|---|---|---|---|---|---|---|---|---|---|---|---|
| B2 | 199 | 1218 | B | 7 | H | 1 | W | 125 | 130 | 59.93 | 18.73 | 0.708529 | 1.71E-05 |
| B3 | 199 | 1218 | B | 7 | H | 3 | W | 75 | 80 | 62.43 | 19.10 | 0.708497 | 1.13E-05 |
| A1 | 199 | 1218 | A | 7 | H | 3 | W | 50 | 55 | 67.29 | 19.67 | 0.708473 | 1.21E-05 |
| A2 | 199 | 1218 | A | 7 | H | 5 | W | 50 | 55 | 70.29 | 20.12 | 0.708448 | 1.71E-05 |
| B4 | 199 | 1218 | B | 8 | H | 3 | W | 100 | 105 | 73.29 | 20.66 | 0.708378 | 1.17E-05 |
| B5 | 199 | 1218 | B | 9 | H | 3 | W | 32 | 37 | 82.07 | 21.50 | 0.708320 | 1.21E-05 |
| B6 | 199 | 1218 | B | 9 | H | 5 | W | 105 | 110 | 85.45 | 21.82 | 0.708309 | 1.71E-05 |
| A4 | 199 | 1218 | A | 9 | H | 4 | W | 82 | 87 | 90.32 | 22.37 | 0.708290 | 1.71E-05 |
| A5 | 199 | 1218 | A | 9 | H | 6 | W | 20 | 25 | 92.67 | 22.65 | 0.708287 | 1.71E-05 |
| B7 | 199 | 1218 | B | 10 | H | 3 | W | 82 | 87 | 94.54 | 22.90 | 0.708248 | 1.50E-05 |
| B8 | 199 | 1218 | B | 10 | H | 4 | W | 87 | 92 | 96.12 | 23.03 | 0.708251 | 1.06E-05 |
| B9 | 199 | 1218 | B | 10 | H | 5 | W | 141 | 146 | 98.17 | 23.17 | 0.708256 | 1.19E-05 |
| B11 | 199 | 1218 | B | 11 | H | 3 | W | 147 | 150 | 107.26 | 24.06 | 0.708199 | 1.71E-05 |
| B12 | 199 | 1218 | B | 12 | H | 3 | W | 107 | 112 | 118.18 | 24.95 | 0.708133 | 1.71E-05 |
| B13 | 199 | 1218 | B | 13 | H | 6 | W | 17 | 22 | 131.57 | 26.03 | 0.708118 | 1.71E-05 |
| 1218A 14H1 15-25cm | 199 | 1218 | A | 14 | H | 1 | W | 15 | 25 | 138.00 | 26.46 | 0.708083 | 1.82E-05 |
| A9 | 199 | 1218 | A | 14 | H | 1 | W | 100 | 105 | 138.73 | 26.52 | 0.708054 | 1.71E-05 |
| 1218A 14H2 115cm | 199 | 1218 | A | 15 | H | 2 | W | 115 | 117 | 140.27 | 26.64 | 0.708054 | 1.84E-05 |
| A10 | 199 | 1218 | A | 14 | H | 3 | W | 80 | 85 | 141.59 | 26.74 | 0.708064 | 1.71E-05 |
| 1218B 15H1 35cm | 199 | 1218 | B | 15 | H | 1 | W | 35 | 38 |  | 27.04 | 0.708070 | 1.86E-05 |
| B14 | 199 | 1218 | B | 15 | H | 2 | W | 30 | 35 | 147.04 | 27.10 | 0.708050 | 1.13E-05 |
| B15 | 199 | 1218 | B | 15 | H | 4 | W | 80 | 85 | 150.56 | 27.34 | 0.708050 | 1.33E-05 |
| 1218B 16H1 25cm | 199 | 1218 | B | 16 | H | 1 | W | 25 | 26.5 | 156.17 | 27.71 | 0.708034 | 1.82E-05 |
| B16 | 199 | 1218 | B | 16 | H | 1 | W | 70 | 75 | 156.63 | 27.75 | 0.708031 | 1.71E-05 |
| B17 | 199 | 1218 | B | 16 | H | 2 | W | 107 | 112 | 158.55 | 27.91 | 0.708029 | 1.21E-05 |
| B18 | 199 | 1218 | B | 16 | H | 5 | W | 60 | 65 | 162.03 | 28.15 | 0.708031 | 1.13E-05 |
| C1 | 199 | 1218 | C | 10 | H | 2 | W | 100 | 105 | 164.77 | 28.42 | 0.708023 | 1.13E-05 |
| 1218C 10H3 5cm | 199 | 1218 | C | 10 | H | 3 | W | 5 | 7 | 165.25 | 28.45 | 0.708019 | 1.82E-05 |
| B19 | 199 | 1218 | B | 17 | H | 3 | W | 120 | 125 | 170.40 | 28.65 | 0.707982 | 1.21E-05 |
| 1218C 11H1 115cm | 199 | 1218 | C | 11 | H | 1 | W | 115 | 117 | 173.77 | 28.91 | 0.708003 | 1.84E-05 |
| C2 | 199 | 1218 | C | 11 | H | 2 | W | 130 | 135 | 175.45 | 29.03 | 0.707986 | 1.21E-05 |
| B20 | 199 | 1218 | B | 18 | H | 5 | W | 10 | 15 | 182.15 | 29.43 | 0.708006 | 1.21E-05 |
| A11 | 199 | 1218 | A | 18 | H | 4 | W | 37 | 45 | 185.68 | 29.66 | 0.707988 | 1.12E-05 |
| A12 | 199 | 1218 | A | 19 | H | 2 | W | 67 | 75 | 192.20 | 30.14 | 0.707976 | 1.71E-05 |
| 1218C 12H6 85cm | 199 | 1218 | C | 12 | X | 6 | W | 85 | 87 | 191.63 | 30.17 | 0.707974 | 1.91E-05 |
| B21 | 199 | 1218 | B | 20 | X | 3 | W | 117 | 122 | 199.97 | 30.73 | 0.707956 | 1.71E-05 |
| B23 | 199 | 1218 | B | 21 | X | 4 | W | 12 | 17 | 211.94 | 31.55 | 0.707930 | 1.50E-05 |






 **Table 2. $^{87}$Sr/$^{86}$Sr data for Site U1406 and the assigned ages and age uncertainties.**

| Site | Hole | Core | Core Type | Section | Section Half | Top Interval (cm) | Bottom Interval (cm) | Depth CSF-A (m) | Depth CCSF-A (m) | $^{87}$Sr/$^{86}$Sr | Internal SE (2σ) | midpoint age assigned (Ma) | lower age (Ma) | upper age (Ma) |
|---|---|---|---|---|---|---|---|---|---|---|---|---|---|---|
| 1406 | A | 2 | H | 4 | w | 89 | 93 | 11.61 | 23.9 | 0.708625 | 0.000018 | 17.60 | 17.45 | 17.75 |
| 1406 | A | 2 | H | 5 | w | 36 | 40 | 12.58 | 24.9 | 0.708607 | 0.000018 | 17.82 | 17.67 | 17.97 |
| 1406 | A | 3 | H | 2 | w | 5 | 8 | 17.14 | 25.7 | 0.708582 | 0.000004 | 18.14 | 17.99 | 18.29 |
| 1406 | A | 3 | H | 4 | w | 89 | 92 | 20.98 | 29.6 | 0.708556 | 0.000004 | 18.43 | 18.28 | 18.58 |
| 1406 | A | 3 | H | 7 | w | 8 | 12 | 24.68 | 33.3 | 0.708556 | 0.000004 | 18.52 | 18.30 | 18.70 |
| 1406 | A | 4 | h | 1 | w | 139 | 143 | 26.61 | 34.7 | 0.708372 | 0.000004 | 20.66 | 20.29 | 20.91 |
| 1406 | A | 4 | H | 3 | w | 81 | 85 | 29.03 | 37.2 | 0.708350 | 0.000004 | 21.00 | 20.60 | 21.30 |
| 1406 | A | 5 | H | 2 | w | 6 | 9 | 36.27 | 45.4 | 0.708323 | 0.000003 | 21.30 | 20.90 | 21.60 |
| 1406 | A | 5 | H | 4 | w | 33 | 37 | 39.55 | 48.7 | 0.708341 | 0.000004 | 21.40 | 21.00 | 21.70 |
| 1406 | A | 6 | H | 6 | w | 74 | 80 | 52.47 | 63.0 | 0.708303 | 0.000004 | 22.20 | 21.80 | 22.50 |
| 1406 | A | 8 | H | 3 | w | 16 | 22 | 66.39 | 77.6 | 0.708267 | 0.000003 | 22.81 | 22.41 | 23.11 |
| 1406 | C | 8 | H | 1 | w | 80 | 84 | | 80.3 | 0.708271 | 0.000014 | 22.90 | 22.50 | 23.20 |
| 1406 | A | 9 | H | 1 | w | 62 | 68 | 73.35 | 84.8 | 0.708252 | 0.000005 | 23.03 | 22.63 | 23.41 |
| 1406 | A | 9 | H | 6 | w | 91 | 95 | 81.16 | 92.6 | 0.708257 | 0.000004 | 23.15 | 22.75 | 23.53 |
| 1406 | B | 10 | H | 3 | w | 140 | 144 | | 95.6 | 0.708249 | 0.000013 | 23.19 | 22.79 | 23.57 |
| 1406 | B | 10 | H | 5 | w | 70 | 74 | | 97.9 | 0.708238 | 0.000013 | 23.32 | 22.92 | 23.69 |
| 1406 | A | 10 | H | 4 | w | 127 | 133 | 88.00 | 101.8 | 0.708208 | 0.000004 | 23.85 | 23.45 | 24.23 |
| 1406 | A | 11 | H | 3 | w | 53 | 59 | 95.26 | 109.2 | 0.708191 | 0.000004 | 24.16 | 23.76 | 24.54 |
| 1406 | A | 12 | H | 1 | w | 90 | 94 | 102.12 | 117.3 | 0.708161 | 0.000006 | 24.72 | 24.32 | 25.09 |
| 1406 | A | 12 | H | 6 | w | 44 | 48 | 109.16 | 124.4 | 0.708142 | 0.000007 | 25.07 | 24.67 | 25.52 |
| 1406 | A | 13 | H | 7 | w | 21 | 25 | 119.45 | 135.0 | 0.708126 | 0.000004 | 25.37 | 24.97 | 25.82 |
| 1406 | A | 14 | H | 4 | w | 97 | 103 | 125.70 | 141.8 | 0.708112 | 0.000007 | 25.64 | 25.24 | 26.09 |
| 1406 | A | 15 | H | 2 | w | 6 | 10 | 131.28 | 148.7 | 0.708097 | 0.000003 | 25.95 | 25.55 | 26.40 |
| 1406 | A | 15 | H | 5 | w | 103 | 107 | 136.81 | 154.3 | 0.708090 | 0.000004 | 26.10 | 25.70 | 26.55 |
| 1406 | A | 16 | H | 4 | w | 26 | 30 | 143.98 | 162.1 | 0.708074 | 0.000004 | 26.37 | 25.97 | 26.82 |
| 1406 | A | 16 | H | 6 | w | 66 | 70 | 147.38 | 165.5 | 0.708074 | 0.000005 | 26.52 | 26.12 | 26.97 |
| 1406 | A | 17 | H | 1 | w | 128 | 132 | 149.40 | 170.1 | 0.708068 | 0.000005 | 26.58 | 26.18 | 27.03 |
| 1406 | A | 17 | H | 4 | w | 86 | 90 | 153.48 | 174.2 | 0.708039 | 0.000003 | 27.31 | 26.71 | 28.21 |
| 1406 | A | 17 | H | 5 | w | 136 | 140 | 155.48 | 176.2 | 0.707999 | 0.000018 | 28.59 | 27.99 | 29.59 |
| 1406 | A | 18 | H | 1 | w | 19 | 23 | 157.51 | 184.7 | 0.707997 | 0.000018 | 28.68 | 28.08 | 29.68 |
| 1406 | A | 18 | H | 2 | w | 94 | 98 | 159.60 | 186.8 | 0.707984 | 0.000018 | 29.24 | 28.64 | 30.24 |
| 1406 | A | 18 | H | 4 | w | 20 | 24 | 161.86 | 189 | 0.707982 | 0.000004 | 29.33 | 28.73 | 30.33 |
| 1406 | A | 18 | H | 5 | w | 96 | 100 | 164.12 | 191 | 0.707976 | 0.000018 | 29.58 | 28.98 | 30.58 |
| 1406 | A | 19 | H | 2 | w | 15 | 19 | 166.37 | 200 | 0.707966 | 0.000018 | 29.97 | 29.37 | 30.97 |



**Table 3. $^{87}$Sr/$^{86}$Sr data for Site 1168 and the assigned ages and age uncertainties.**

| Site | Hole | Core | Core Type | Section | Section Half | Top Interval (cm) | Bottom Interval (cm) | Depth (mbsf) | $^{87}$Sr/$^{86}$Sr | Internal SE (2σ) | central age (Ma) | min age (Ma) | max age (Ma) | |
|------|------|------|-----------|---------|--------------|-------------------|----------------------|--------------|---------------------|------------------|------------------|--------------|--------------|--|
| 1168 | A | 30 | X | 5 | W | 2 | 8 | 278.25 | 0.708724 | 1.82E-05 | **16.29** | 16.14 | 16.44 | |
| 1168 | A | 33 | X | 2 | W | 52 | 58 | 302.75 | 0.7085492 | 1.82E-05 | **18.57** | 18.35 | 18.74 | |
| 1168 | A | 37 | X | 1 | W | 43 | 49 | 339.26 | 0.7084685 | 1.82E-05 | **19.67** | 19.39 | 19.87 | |
| 1168 | A | 40 | X | 3 | W | 59.5 | 64.5 | 371.22 | 0.7083942 | 1.82E-05 | **20.55** | 20.18 | 20.80 | |
| 1168 | A | 41 | X | 1 | W | 5 | 7.5 | 377.36 | 0.7083771 | 1.41E-05 | **21.02** | 20.62 | 21.32 | |
| 1168 | A | 41 | X | 1 | W | 135 | 138 | 378.66 | 0.7083633 | 1.35E-05 | **21.24** | 20.84 | 21.54 | |
| 1168 | A | 41 | X | 6 | W | 34 | 36.5 | 385.15 | 0.7083596 | 1.35E-05 | **21.28** | 20.88 | 21.58 | |
| 1168 | A | 41 | X | 7 | W | 34 | 47.5 | 386.46 | 0.7083715 | 1.41E-05 | **21.33** | 20.93 | 21.63 | |
| 1168 | A | 42 | X | 3 | W | 45 | 25.5 | 390.78 | 0.7083377 | 1.62E-05 | **21.63** | 21.23 | 21.93 | |
| 1168 | A | 42 | X | 4 | W | 137 | 139 | 392.78 | 0.7082875 | 1.76E-05 | **22.47** | 22.07 | 22.77 | |
| 1168 | A | 43 | X | 5 | W | 55 | 57 | 403.06 | 0.7082624 | 1.95E-05 | **22.90** | 22.50 | 23.28 | |
| 1168 | A | 44 | X | 2 | W | 69 | 71 | 408.3 | 0.7082843 | 1.95E-05 | **23.20** | 22.80 | 23.58 | |
| 1168 | A | 45 | X | 3 | W | 143 | 145 | 420.14 | 0.708272 | 1.92E-05 | **23.30** | 22.90 | 23.68 | |
| 1168 | A | 46 | X | 1 | W | 49 | 55 | 425.82 | 0.7082296 | 1.82E-05 | **23.46** | 23.06 | 23.84 | |
| 1168 | A | 48 | X | 4 | W | 65 | 67.5 | 449.65 | 0.708231 | 1.95E-05 | **23.70** | 23.30 | 24.08 | |
| 1168 | A | 51 | X | 4 | W | 38 | 42 | 478.2 | 0.7081923 | 1.82E-05 | **24.30** | 23.90 | 24.68 | |
| 1168 | A | 55 | X | 5 | W | 33 | 39 | 518.06 | 0.7081716 | 1.82E-05 | **24.51** | 24.11 | 24.96 | |
| 1168 | A | 56 | X | 5 | W | 37 | 43 | 527.8 | 0.7081292 | 1.82E-05 | **25.00** | 24.60 | 25.45 | |
| 1168 | A | 57 | X | 5 | W | 113 | 117 | 538.25 | 0.7081581 | 1.86E-05 | **25.20** | 24.80 | 25.65 | |
| 1168 | A | 59 | X | 4 | W | 83 | 87 | 555.75 | 0.7080568 | 2.19E-05 | **26.84** | 26.34 | 27.44 | |
| 1168 | A | 60 | X | 2 | W | 57 | 61 | 562.09 | 0.7080503 | 1.84E-05 | **27.02** | 26.52 | 27.62 | |

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
