# Peer review of "Nonlinear increase in seawater 87Sr/86Sr in the Oligocene to"

_Climate of the Past, 2023_

## Author Comment (AC1)

We thank the reviewer for the attentive reading and many constructive suggestions to clarify the paper and correct details of presentation. Author response to anonymous Reviewer comment. Reviewer comments in black, response in violet and quotations from revised text in blue.

**Reviewer comment**

In "Nonlinear increase in seawater 87Sr/86Sr in the Oligocene to early Miocene and implications for climate-sensitive weathering" Stoll et al., present newly generated 87Sr/86Sr data for (I)ODP Sites 1168, 1218, and U1406. Especially the work on Site 1218 is very valuable in my opinion, because Sr ratios can be directly compared to the benthic foraminiferal d18O climate/cryosphere proxy from the same site. The authors find that the gradual increase in 87Sr/86Sr ratios stalls or even decreases when d18O is high and that the increase in 87Sr/86Sr accelerates when d18O is low. These findings are interpreted to reflect ITCZ related and/or Antarctic weathering related controls on Sr transport to the oceans.

The paper is well written, and well-illustrated. I am not fully convinced by the ITCZ hypothesis, but it can stay in the paper. I have one major comment on the use of the CCSF-A scale for U1406. This would require the authors to transpose the biostratigraphy from the Site U1406 CSF-A to CCSF-A depths, but I think that is worth the effort. Overall, I strongly recommend this manuscript for publication, and would like to congratulate the authors with their nice study.

Major comment:

• L93. Extraordinary effort has gone into generating the best available depth model for Site U1406, and I strongly advise to use the composite CCSF-A scale, and against using the Hole A specific CSF-A scale. There is no benefit to it, and mixing depth models only serves to confuse the reader.

We thank the reviewer for the suggestion to emphasize the composite CCSF-A scale. The CCSF-A correlates the sites based on physical properties and trace element ratios from XRF scanning. As detailed in van Peer et al (2017), in some sections, there are gaps or condensed intervals in one hole but not the other, and the XRF scanning data has formed the basis of one possible correlation. The composite scale is likely to continue to evolve as additional data is published. For example, additional data (currently in review) including benthic foraminiferal d18O and bulk carbonate d18O highlights some intervals where the CCSF-A scale requires revision for off-splice holes. In light of this, in showing age-depth plots to indicate the position of hiatus', we used the CSF-A because the samples bracketing hiatuses in our study are only from Hole A, and the hiatus duration is unambiguous in this hole. We offered maximum transparency by providing both the Hole A specific CSF-A scale and the proposed CCSF-A equivalent, in Table 2. This will allow future readers to update our presented data to any future composite depth scales. The composite scale was used in Figure 2. In the meantime, we can agree to plot the remaining figure 4d (the only one still presenting CSF-A from 1406) on CCSF-A and entrust future users of the data to employ due caution when extrapolating results to Hole B and C.

[Figure]

Some smaller comments, which the authors may address at their discretion:

- General comment: Perhaps add a little map with site locations?

We appreciate this suggestion and propose to include a small map with site locations in the revision.

- L27. Perhaps state where ODP Site 1168 and IODP Site 1406 are from?

We thank the reviewer for this suggestion to add a map and describe in the introduction text the locations, we propose to implement both suggestions. The proposed map would be similar to this one: .

[Figure]

Location of sites investigated in this study. Reconstructions using the plate tectonic reconstruction service ODSN (www.odsn.de).

- L55. 4 million years, seems like a very large uncertainty, even for biostratigraphic age control from the 1980s. Million is abbreviated by a big M.

We understand the reviewers impression that 4 million years is a large uncertainty for biostratigraphic age models. Nonetheless, this uncertainty in biostratigraphy is exactly as described in the cited article of Miller et al 1998 and we retain it as originally cited. In this sentence we are reporting a duration of time not an age; to our, million years as a duration is elsewhere abbreviated m.y. and is so abbreviated in the Miller et al., 1988 references. We acknowledge that this is distinct from an absolute age where capital M is used. To avoid confusion, in the revision we propose to write out the duration as million years.

• L55. I would say "astrochronology" instead of "cyclostratigraphy" in this context. The former refers to, in my mind, the alignment of cyclic stratigraphic records (in the depth domain) to an astronomical solution (in the age domain). The latter is, as I understand it, thus a description of cyclically alternating lithologies/proxy records in the stratigraphic depth domain.

We thank the reviewer for the suggestion and propose to update to the term astrochronology.

• L63. Please make sure the Westerhold et al., 2020 age model is used. At 26.5 Ma a hundred-thousand-year correction of the Pälike et al., 2006 ages is included in Westerhold et al., 2020.

We thank the reviewer for prompting us to clarify this. The tables and figures do indeed employ the Westerhold 2020 age model, and the Westerhold age model was already specified as the update in section 2.1 in Methods. In the introduction we will reword this sentence to make the point earlier in the text.

• L71. van Peer et al. 2017a, shows this hiatus in the age domain. van Peer is with a small v.

We thank the reviewer for the suggestion. We will update the reference with the capitalization error which occurred during reference import from online bibliographical data. We adjust the phrasing in the introduction as follows:

At site 1406, Sr isotope stratigraphy improves constraints on the duration of an early Miocene hiatus (Norris et al., 2014; van Peer et al., 2017).

• L79. Astrochronologic. (See previous comment). (we will adjust, as described above)

• L81. Delete "GTS". The Westerhold et al., 2020 CENOGRID record is separate from the GTS2020. (NB: The G(P)TS2020 should not be used, due to incorrect incorporation of several chron boundary reversal ages. Most notably the Liebrand et al., 2016 ages, which were not all being considered reliable or deviant from GTS2012 outside of error, their Supp. Fig. S6.)

We thank the reviewer for prompting us to clarify this, we delete the reference to GTS 2020 in the text.

• L101. This is an "old" ODP site that probably used the mbsf terminology, instead of CSF-A scale. Perhaps clarify this point in the text?
We thank the reviewer for prompting us to clarify this, which we will implement in the text and figure axes.

[Figure]

- L120/121. Sigma and SD are used. Perhaps standardize?  Yes, we will standardize this.

- L135-137. This is the same hiatus identified in van Peer et al., 2017a.

We thank the reviewer for prompting us to add this information in the discussion and not only in the introduction as noted previously.  We add the following sentence to the discussion (first sentence in section 4.2.1), rather than the results section which focuses on a brief summary of the new results.

A condensed interval and hiatus have been recognized in the Oligocene to early Miocene sediments of U1406 on the basis of bio- and magnetostratigraphy (Figure 4 ) (Norris et al., 2014; Van Peer et al., 2017).

- L147. This definition of the MOGI deviates from the original (Liebrand et al., 2017), which is defined as lasting from 28.0 to 26.3 Ma. Perhaps explain why a longer lasting MOGI is defined. I have no problem with a longer lasting MOGI, it is just an arbitrary definition, but still good to be clear how and why it differs.

We thank the reviewer for prompting us to clarify the basis for the duration illustrated here, which we propose to implement in the caption of Figure 1.

We highlight this duration of MOGI on the basis of the 1218 benthic $\delta^{18}O$ record as indicated in Figure 3; it is slightly longer than the 28 to 26.3 Ma MOGI defined by Liebrand et al., 2017.

- L173. Is an inverted increase not a decrease?

We thank the reviewer for the suggestion to simplify the language and describe this as a decrease, we will adjust the text accordingly.

- L180. In Fig. 3a and b this looks like a decrease, not a reduction in the increase. Confusing language.
We thank the reviewer for the suggestion to simplify the language and describe this as a decrease, we will adjust the text accordingly.

- L181. I think it is important to redefine the MOGI as used in this study. (See previous comment) (adjusted as described in response to previous comment).

- L183. "Warming" refers to deep-sea (and potentially high latitudes). Perhaps clarify.

We thank the reviewer for prompting us to clarify this, we agree and will implement this in the text.

- L183-184. Also looks like a modest decrease, not a "slowing of increase".

We agree and will update the description used in the text.

• L186. Looks like a decrease to me. We agree, but for a conservative interpretation of the error bars we indicate it as a stabilization or a decrease.

• L212-L215. I do not understand this sentence.

We propose to rewrite the sentences for improved clarity as below:

However, climatically-driven changes in the position of main heavy rainfall belts such as ITCZ is usually limited to < 10 degrees latitude and may be longitudinally variable (Atwood et al., 2020). A movement of precipitation belts would have a significant consequence on global riverine $^{87}Sr/^{86}Sr$ only in cases of fortuitous distribution of bedrock of widely different ages across the length scale of ITCZ movement. If a northward shift of the mean ITCZ significantly increased the Sr flux from a region of nonradiogenic Sr, the marine $^{87}Sr/^{86}Sr$ could experience a transient decrease. One potential such configuration could be the exposure of highly weatherable nonradiogenic rocks of the Deccan volcanic series of India and the Ethiopian Traps, located just north of the equator in the late Oligocene(Kent and Muttoni, 2013).

• L241-244. This finding of Oslick can now be reflected on using the new data and age model. Just stating it seems a bit odd. To me it seems that the increase in Sr ratios follows δ18O maxima much quicker, compared to the 1 M.y. lag Oslick found.

We agree with this comment and propose to add the following sentence at the end of this paragraph.

With higher resolution benthic $\delta^{18}O$ from 1218, we resolve more rapid responses of the $^{87}Sr/^{86}Sr$ ratio to several deglaciation phases.

• L279-282. Independent Sr age control is great. However, I would mention–also in writing–that for Site U1406 a magnetostratigraphic age model is publicly available (van Peer, 2017a). Are Sr and PMAG age models for Site U1406 in agreement with one another? Or are there any large discrepancies?

We thank the reviewer for prompting us to textually comment on the comparison of PMAG and Sr isotope stratigraphy, which were illustrated in Figure 4d. We propose to add this phrase :

Between 26.4 and 21 Ma, the $^{87}Sr/^{86}Sr$ age model is in close agreement with that devised from magnetostratigraphy(Van Peer et al., 2017).

The Sr isotopes provide an additional constraint on the early Miocene section above the hiatus for which no PMAG tiepoints are published.

• L305. Middle Miocene (25 to 27 Ma). Do the authors mean Oligocene?

Yes, this should state Middle Oligocene, and will be corrected in the revised text.

• L353. LP = LDP?. Who is PG?

We have adjusted the initials in the acknowledgments for clarity.

References in the response

Hodell, David A., Gregory A. Mead, and Paul A. Mueller. "Variation in the strontium isotopic composition of seawater (8 Ma to present): Implications for chemical weathering rates and dissolved fluxes to the oceans." *Chemical Geology: Isotope Geoscience section* 80, no. 4 (1990): 291-307.

Paytan, Adina, Elizabeth M. Griffith, Anton Eisenhauer, Mathis P. Hain, Klaus Wallmann, and Andrew Ridgwell. "A 35-million-year record of seawater stable Sr isotopes reveals a fluctuating global carbon cycle." *Science* 371, no. 6536 (2021): 1346-1350.

Atwood, A. R., Donohoe, A., Battisti, D. S., Liu, X., and Pausata, F. S.: Robust longitudinally variable responses of the ITCZ to a myriad of climate forcings, Geophysical Research Letters, 47, e2020GL088833, doi.org/10.1029/2020GL088833, 2020.
Kent, D. V. and Muttoni, G.: Modulation of Late Cretaceous and Cenozoic climate by variable drawdown of atmospheric pCO 2 from weathering of basaltic provinces on continents drifting through the equatorial humid belt, Climate of the Past, 9, 525-546, doi.org/10.5194/cp-9-525-2013, 2013.
Norris, R., Wilson, P., and Blum, P.: Proceedings of the Integrated Ocean Drilling Program Exp. 342, College Station, TX: Integrated Ocean Drilling Program, doi.org/10.2204/iodp.proc.342.107.2014, 2014.
van Peer, T. E., Xuan, C., Lippert, P. C., Liebrand, D., Agnini, C., and Wilson, P. A.: Extracting a detailed magnetostratigraphy from weakly magnetized, Oligocene to early Miocene sediment drifts recovered at IODP Site U1406 (Newfoundland margin, northwest Atlantic Ocean), Geochemistry, Geophysics, Geosystems, 18, 3910-3928, doi.org/10.1002/2017GC007185, 2017.

---

## Author Comment (AC3)

We thank Prof. Derry for the attentive reading and many constructive suggestions to clarify the paper including a deeper representation of the contrasting ocean response times of Sr isotopes and oxygen isotopes. Author response to community comment by L. Derry. Reviewer comments in black, response in violet and quotations from revised text in blue.

**CC**

This ms presents new 87Sr/86Sr data from the period of rapid rise in in the isotopic composition of Sr dung the late Oligocene to early Miocene. The innovation is using a common orbitally tuned age model (CENOGRID) for the ODP core samples. The improved precision of the CENOGRID age model allows the authors to recognize additional "fine" structure in the overall increasing 87Sr/86S6 signal through interval from about 32 to 19 Ma. The apparently higher sedimentation rate at Site 1218 allows more detail to emerge that at sites U1406 and 1168. Nevertheless correlation with the Sr isotope data improves the age model for those sites as well. It also provides an improved correlation between the Sr and benthic oxygen isotope records. All this seems to be well documented and straightforward, and is a useful refinement. It should be possible to extend this approach to the high resolution data in Oslick 1994, for example.

It would be helpful to have lat-lon-depth at minimum for the three sites. Most of us don't have a ODP site map in our head.

We appreciate the suggestion to provide a site map also made by reviewer 1 and will add this to the revision. The latitude-longitude coordinates and depths for the three ODP sites were provided in the methods subsection titled Sediments.

[Figure]

Location of sites investigated in this study. Reconstructions using the plate tectonic reconstruction service ODSN (www.odsn.de).

There is a sedimentation rate curve for U1406 and 1168 – why not also 1218?

We have provided sedimentation rate plots for the sites in which a new age model is provided here via Sr isotope stratigraphy (U1406 and 1168). We do not illustrate the sedimentation rate for the site for which we employ a previously published age model based on astrochronology (ODP 1218). The sedimentation rate from ODP 1218 was published in Supplementary Figure S2 in Palike et al., 2006.

While not necessary here, the uncertainties on 87Sr/86Sr could be improved by up to a factor of 2 by using modern TIMS in place of ICPMS.

We thank the Prof. Derry for pointing out the higher precision of TIMS which may be useful to consider in future studies.

There is some discussion of the potential link between Antarctic glaciation and variations in SW 87Sr/86Sr based on the observation that changes in the rate of 87Sr/86Sr are related to climate events recorded in d18O.  It's much harder to constrain the drivers of 87Sr/86Sr in this interval as there are multiple potential drives.  The apparent correlation with benthic d18O signal could reflect glacial influence on weathering in Antarctic as they propose, and they also note that climate shifts could influence the position of the ITCZ and this impact rainfall pattern in areas that could change the Sr flux or its isotope composition. Changes in precipitation (and glaciation in high altitude zones) could also occur the beyond the polar and  ITCZ-impacted regions. A further potential player is the unroofing of highly radiogenic sources rocks in the Himalaya at this time (Galy et al 1999 *Tectonophysics*).  The detailed timing and impact of this unroofing is not known with sufficient accuracy to assign to particular changes in slope occurring in < 1 Myr reported here, but may well be playing a major role on the overall rapid increase (Myrow et al 2016 *EPSL*).  There are also known changes in the 87/86 ratio of rivers draing NE Tibet in this interval (Yang et al. 2022 *GSAB*).  The combined range of potential climate and tectonic/geologic factors makes it difficult to move beyond a somewhat speculative statement here.  While there is nothing "wrong" about the discussion of possible contributions from radiogenic rocks in Antarctica, or changes in weathering of the Deccan flood basalts, or other mechanisms, there is nothing particularly convincing about any of this either.

We appreciate the suggestion of additional references relating to the tectonic processes affecting the exposure of highly radiogenic bedrocks which may contribute to the overall rapid increase in the 87Sr/86Sr ratio of the Oligocene, and propose to include these in the introduction.

It's worth keeping in mind that the oceanic residence time of Sr is ≈ 3 Myr, much longer than for O, so the response time scales are quite different (e.g. Richter and Turekian, 1993).  If d18O is responding to a climate driver the 87Sr/86Sr response will be lagged and damped significantly. It's not that easy to get d(87/86)/dt rates up to 2x10$^{-4}$ per Myr as they propose for an interval just after 29 Ma, that would require large and rapid fluctuations in the input of Sr and it's isotopic composition. Before taking these short term variations as gospel it would be worth doing a calculation to see what kind of weathering input variations are necessary to drive such sharp changes.

We appreciate Prof. Derry's suggestion to illustrate more clearly the implications of different ocean response times of oxygen isotopes and Sr isotopes (eg the attenuation of the input forcing by the long residence time of Sr), which we had previously mentioned only in the text.   We fully agree that rapid short term changes in the ocean 87Sr/86Sr ratio require large changes in fluxes and or isotopic compositions. An ocean box model can be used to simulate the effect of the long residence time on the phasing of the forcing.  As Prof. Derry mentioned previously, the Sr cycle of the Oligocene is underconstrained, so we do not propose that a unique inverse solution (e.g. weathering flux or its isotopic ratio) can be attained from the present data. As an example, we propose to illustrate in a supplemental figure, for a 2.5 Ma residence time (Hodell et al., 1990), the required changes in isotopic composition of the Sr influx to the ocean which

would be needed to match the observed high frequency changes in the ocean 87Sr/86Sr ratio.  We note that some studies suggest a shorter residence time in the Oligocene (Paytan et al., 2021), which would require less extreme forcing to simulate the variations. We hope that these examples elucidate more clearly the relationship between Sr isotope forcing and the climate variations and stimulate future work.

We propose to add three sentences to the discussion of this process (new sentences in bold):

In either case, the long residence time of Sr in the ocean would result in lags between onset of elevated fluxes and peak response in ocean chemistry and would cause significant attenuation of the time-varying input signal. **An example of the phasing and amplitude variation in the $^{87}$Sr/$^{86}$Sr of the Sr influx which could yield the observed trends in marine $^{87}$Sr/$^{86}$Sr is illustrated in Supplementary Figure 1, for a sample residence time of 2.5 million years as suggested by Hodell et al., 1990. A shorter residence time has been proposed for the Oligocene by Paytan et al., (2021); for a shorter residence time, a less extreme forcing would be required to simulate our observations.  We caution that because the Sr isotopic system of the Oligocene to Early Miocene is underconstrained, the observations of oceanic $^{87}$Sr/$^{86}$Sr do not provide a unique solution for the variation in fluxes and/or their isotopic composition.**

[Figure]

**Supplementary Figure illustrating the response time of variation in Sr influxes in relation to benthic δ¹⁸O. a) Measured Site 1218 ⁸⁷Sr/⁸⁶Sr (symbols) and the smoothed fit from local linear regression (blue line) as well as a model fit to the curve (dashed red line) forced as illustrated in panel d). b) the derivative of the smoothed fit (black line) and the slope of each linear segment (purple square), together with the 1σ uncertainty on the slope (vertical error bar, 68% confidence interval) and the age range of the local linear fit (horizontal bar). Shading indicates sectors in which ⁸⁷Sr/⁸⁶Sr rises more slowly than the average rate over 32 to 18 Ma. c) Modeled changes in the Sr isotope ratio of the influx albe to generate the red dashed curve illustrated in panel a) using a single ocean box with a residence time of 2.5 million years (Hodell et al., 1990) for an oceanic Sr concentration of 87μM, and constant influx and outflux. d) Benthic δ¹⁸O measurements (gray points) and lines showing 20 point running mean, as illustrated in Figure 3, from (Pälike et al., 2006) (Holbourn et al., 2015) and (Westerhold et al., 2020). All data are plotted on the orbitally tuned CENOGRID timescale(Westerhold et al., 2020) with shading and time intervals of interest as in Figure 3.**

Overall this is a paper that shod be published with modest changes. In my view the most important changes needed are to take a more cautious approach to proposing drivers for the short term isotopic variability.

Louis Derry

References in the response

Hodell, David A., Gregory A. Mead, and Paul A. Mueller. "Variation in the strontium isotopic composition of seawater (8 Ma to present): Implications for chemical weathering rates and dissolved fluxes to the oceans." *Chemical Geology: Isotope Geoscience section* 80, no. 4 (1990): 291-307.

Paytan, Adina, Elizabeth M. Griffith, Anton Eisenhauer, Mathis P. Hain, Klaus Wallmann, and Andrew Ridgwell. "A 35-million-year record of seawater stable Sr isotopes reveals a fluctuating global carbon cycle." *Science* 371, no. 6536 (2021): 1346-1350.

Holbourn, A., Kuhnt, W., Kochhann, K. G., Andersen, N., and Sebastian Meier, K.: Global perturbation of the carbon cycle at the onset of the Miocene Climatic Optimum, Geology, 43, 123-126, doi.org/10.1130/G36317.1, 2015.
Pälike, H., Norris, R. D., Herrle, J. O., Wilson, P. A., Coxall, H. K., Lear, C. H., Shackleton, N. J., Tripati, A. K., and Wade, B. S.: The heartbeat of the Oligocene climate system, science, 314, 1894-1898, DOI: 10.1126/science.1133822, 2006.
Westerhold, T., Marwan, N., Drury, A. J., Liebrand, D., Agnini, C., Anagnostou, E., Barnet, J. S., Bohaty, S. M., De Vleeschouwer, D., and Florindo, F.: An astronomically dated record of Earth's climate and its predictability over the last 66 million years, Science, 369, 1383-1387, DOI: 10.1126/science.aba685, 2020.